# Mildew Resistance Locus O Genes *CsMLO1* and *CsMLO2* Are Negative Modulators of the *Cucumis sativus* Defense Response to *Corynespora cassiicola*

**DOI:** 10.3390/ijms20194793

**Published:** 2019-09-26

**Authors:** Guangchao Yu, Qiumin Chen, Xiangyu Wang, Xiangnan Meng, Yang Yu, Haiyan Fan, Na Cui

**Affiliations:** 1College of Horticulture, Shenyang Agricultural University, Shenyang 110866, China; yugc7674@163.com (G.Y.); QiuminChen2019@163.com (Q.C.); wxykids@163.com (X.W.); 2College of Bioscience and Biotechnology, Shenyang Agricultural University, Shenyang 110866, China; nouis0820@126.com (X.M.); yy7603@163.com (Y.Y.); 3Key Laboratory of Protected Horticulture of Ministry of Education, Shenyang Agricultural University, Shenyang 110866, China

**Keywords:** *Cucumis sativus*, CsMLO1 gene, *CsMLO2* gene, plant defense, expression analysis, *Corynespora cassiicola*

## Abstract

Corynespora leaf spot caused by *Corynespora cassiicola* is one of the major diseases in cucumber (*Cucumis sativus* L.). However, the resistance mechanisms and signals of cucumber to *C. cassiicola* are unclear. Here, we report that the mildew resistance locus O (MLO) genes, *CsMLO1* and *CsMLO2*, are both negative modulators of the cucumber defense response to *C. cassiicola*. Subcellular localization analysis showed that CsMLO1 and CsMLO2 are localized in the plasma membrane. Expression analysis indicated that the transcript levels of *CsMLO1* and *CsMLO2* are linked to the defense response to *C. cassiicola*. Transient overexpression of either *CsMLO1* or *CsMLO2* in cucumber cotyledons reduced resistance to *C. cassiicola*, whereas silencing of either *CsMLO1* or *CsMLO2* enhanced resistance to *C. cassiicola*. The relationships of pathogenesis-related proteins, reactive oxygen species (ROS)-associated genes, and abscisic acid (ABA)-related genes to the overexpression and silencing of *CsMLO1*/*CsMLO2* in non-infested cucumber plants were investigated. The results indicated that *CsMLO1* mediated resistance against *C. cassiicola* by regulating the expression of pathogenesis-related proteins and ROS-associated genes, as well as through ABA signaling pathway-associated genes. The *CsMLO2*-mediated resistance against *C. cassiicola* primarily involves regulation of the expression of pathogenesis-related proteins. Our findings will guide strategies to enhance the resistance of cucumber to corynespora leaf spot.

## 1. Introduction

Cucumber target leaf spot (TLS) is caused by *Corynespora cassiicola* (Berk and Curt, Wei) and has resulted in increasing losses in cucumber production in recent years. Thus far, research on TLS has primarily focused on identification, control methods and biological characteristics. Several resistant varieties, including Jinyou 3 [1], Jinyou38 [1], Marketmore97 [2], HygBrid72502 [3], and USDA6623E [4], have been identified by disease resistance phenotypes in both Asia and Europe. At present, the inheritance of TLS resistance in cucumber has been found in several studies by using marker-assisted selection (MAS) and simple sequence repeat (SSR) techniques. A single dominant gene, *Cca*, show TLS resistance in the Royal Sluis 72502 cucumber line [5], while the single recessive genes *cca-1*, *cca-2* and *cca-3* show resistance to cucumber TLS [1,6,7]. Therefore, the controversy about the genetic control of TLS resistance, and whether TLS resistance is controlled by a single dominant or recessive gene remain unclear. In addition, the genetic and physical distances of linked markers from these studies are still far away from the target gene. These markers are not breeder-friendly and not amendable for high-throughput genotyping. Most plants have non-host resistance to microbial invaders, and this resistance mechanism provides broad-spectrum and strong resistance to non-adapted pathogens in plants [8]. The mildew resistance locus O (MLO) gene in non-host defense plays an important role in the model system of the interaction of barley with *Blumeria graminis* f. sp. *hordei* (*Bgh*) [9,10]. Inducible defense responses in nonhost plants is mainly caused by the hypersensitive response (HR) of host plants, including the accumulation of ROS, the activation of pathogenesis-related genes, and localized reinforcement of the plant cell walls [11,12]. Therefore, we can screen related resistance genes and study their molecular mechanisms of host resistance to cucumber TLS. Elucidating the molecular mechanism of cucumber-*C. cassiicola* interactions is important for establishing ideal varieties.

MLO genes are considered negative regulators of plant immunity [13]. For instance, *Arabidopsis* MLO2, MLO6 and MLO12 have genetic redundancy in which *Atmlo2 Atmlo6 Atmlo12* triple mutants are fully resistant to the powdery mildew pathogen and *Atmlo2* single mutants are partially resistant [14,15,16]. Subsequent studies revealed that leaf cells show spontaneous cell death, callose deposition and reactive oxygen species (ROS) accumulation in the genetic background of *Atmlo2 Atmlo6 Atmlo12* triple mutants and *Atmlo2* single mutants [17]. In addition, after inoculation with the bacterial pathogen *Pseudomonas syringae*, induction of *Arabidopsis MLO2* gene expression is associated with salicylic acid (SA) signaling, and systemically enhanced in plant leaves exhibiting systemic acquired resistance (SAR) [18]. Furthermore, it may be worth mentioning that *Arabidopsis mlo2* mutants are defective in SAR against *P. syringae* [18]. At present, *MLO* genes have been implicated in other resistance pathways. Barley *mlo* alleles enhance susceptibility to the necrosis-inducing toxin from *Bipolaris sorokiniana* [19]. Studies have identified the interactions between *Arabidopsis* and bacteria (e.g., *Pseudomonas syringae*), *Arabidopsis* and oomycetes (e.g., *Hyaloperonospora arabidopsidis* and *Albugo* spp.), and *Arabidopsis* and fungus (*Colletotrichum higginsianum*) [20,21,22,23]. Furthermore, research has recently shown that the triple mutant line *mlo2-5 mlo6-2 mlo12-1* increased susceptibility to *P. syringae* pv. *Maculicola* [24]. In our laboratory, transcriptome and iTRAQ analyses identified two CsMLOs of cucumber that are involved in the response to *C. cassiicola* [25,26]. However, the CsMLOs function in *C. cassiicola* infection, and the defense mechanisms have not been described in cucumber.

Plant self-resistance causes a series of intracellular signaling responses under stress, and they include physiological and biochemical processes. ROS are important signal transduction molecules involved in different cellular signaling pathways [27,28]. The main sources of ROS production include superoxide radicals (O_2_·^−^), hydroxyl radicals (OH), hydrogen peroxide (H_2_O_2_) and singlet oxygen [29,30]. In plant immunity, ROS production in response to pathogen recognition is mediated by the In plant immunity, ROS production in response to pathogen recognition is mediated by the oxidase nicotinamide adenine dinucleotide phosphate (NADPH) oxidases, also called respiratory burst oxidase homologs (RBOHs), and *arabidopsis* AtRBOHD and AtRBOHF plays a key role in apoplastic ROS generation during plant-pathogen interactions [31,32]. Moreover, the outbreak of ROS, cell death that is phenotypically similar to a hypersensitive response (HR) and increased levels of ROS-producing enzymes play an active role in plant defense [33,34]. Additionally, stoma closure participates in H_2_O_2_ production in *Arabidopsis*, which is promoted in guard cells by pathogen-associated molecular patterns (PAMPs)-induced abscisic acid (ABA) signaling, and this pattern increases plant resistance to pathogens [35,36,37]. Therefore, ABA is involved in the development of defense responses during plant-pathogen interactions to block pathogen invasion. Three core components include the ABA receptors PYLs, the negative regulator type 2C protein phosphatases (PP2Cs) and positive regulator subfamily 2 of SNF1-related kinases (SnRK2s) in ABA signal transduction plants [38]. ABA-insensitive 5 (ABI5), as a basic leucine-zipper (bZIP) transcription factor, is essential to ABA-mediated developmental checkpoints [39]. In cucumber seedlings, *CsPYL2*, *CsPP2C2* and *CsSnKR2.2* respond to the ABA signaling pathway under drought stress [38]. Furthermore, pathogenesis-related proteins (PR proteins) proteins, including pathogenesis-related protein 1-1a (*PR1-1a*), β-1, 3-glucanase (*PR2*) and chitinase (*PR3*), are increased to promote resistance to pathogens [40,41]. However, these aspects of ROS and ABA signal transduction remain largely unexplored in *CsMLOs-C. cassiicola* interactions.

In this study, *CsMLO1* (XM_004148737) and *CsMLO2* (XM_004142345) were shown to be induced by *C. cassiicola*. A subcellular localization analysis showed that these proteins are located on the plasma membrane. *CsMLO1* and *CsMLO2* respond differently to abiotic stresses. An investigation of their defense roles via transient *CsMLO1/CsMLO2*-silencing and *CsMLO1/CsMLO2*-overexpressing cucumber plants showed that *CsMLO1* and *CsMLO2* both function as negative modulators in the host resistance response to *C. cassiicola*. Furthermore, *CsMLO1* silencing might participate in the defense response to *C. cassiicola* by inducing the significant upregulation of several ROS-associated genes, ABA signaling pathway-associated genes or pathogenesis-related proteins. We concluded that *CsMLO1* participated in the defense response to *C. cassiicola* in plants through multiple signaling regulatory networks. *CsMLO2* regulated the expression of pathogenesis-related proteins in uninfected cucumbers. There was no pattern in the expression of ROS- and ABA-related genes in *CsMLO2* transgenic cucumbers, suggesting that this gene might regulate other signaling pathways that affect cucumber resistance to *C. cassiicola*.

## 2. Results

### 2.1. Phenotypic Characterization of Cucumber Cultivars after C. cassiicola Infection

All cucumber cultivars were inoculated on the second leaves with *C. cassiicola* and infected after 5 days. The disease index statistics of each strain are shown in Table 1, which shows that there were obvious differences in disease resistance among different cucumber strains. Jinyou 38 was highly resistant to *C. cassiicola*. B21-a-2-1-2 and F10 were moderately resistant to *C. cassiicola*. B21-a-2-2-2, 995, Jinyan 4 and Xintaimici were susceptible to *C. cassiicola*. Based on the disease index survey, we selected the high-resistance variety Jinyou 38 and the susceptible variety Xintaimici to carry out the next experiments [42].

### 2.2. Isolation and Phylogenetic Analysis of CsMLO1 and CsMLO2

A search of the cucumber genomics database (http://cucurbitgenomics.org/) and gene prediction analysis of the cucumber genome led to the identification of two hypothetical full-length *CsMLO1* and *CsMLO2* transcripts in the Xintaimici and Jinyou 38 cultivars. The 1749 bp full-length *CsMLO1* cDNA encoding a 66.92 kDa protein with 582 amino acid residues and the 1725 bp full-length *CsMLO2* cDNA encoding a 64.25 kDa protein with 574 amino acid residues were identified from cucumber leaves (Appendix A). The cDNA sequences of *CsMLO1* were cloned from the experimental materials of Jinyou 38 and Xintaimici, and the results showed that there was no difference in cDNA sequences of *CsMLO1* between the Jinyou 38 variety and Xintaimici variety. Similarly, similar cloning and alignment methods found that cDNA sequences of *CsMLO2* also had no differences between the Jinyou 38 variety and Xintaimici variety. A sequence analysis predicted that CsMLO1 and CsMLO2 were homologous to MLO proteins. Both homoeologous proteins shared high similarity with proteins for powdery mildew fungus susceptibility in pepper, *Arabidopsis*, turnip and barley. These amino acid sequences were highly conserved and contained seven transmembrane structures (TM1–TM7) and a C-terminal calmodulin-binding domain (CaMBD) in the C-terminal region (Appendix A).

A phylogenetic tree containing 21 MLO proteins from dicots and monocots demonstrated the phylogenetic relationships. CsMLO1 and CsMLO2 proteins were clustered with the five dicot subfamilies containing AtMLO2, AtMLO6, AtMLO12, BrMLO1, LeMLO1, and CaMLO1 and the monocot OsMLO1 (Appendix A). Each of these proteins was previously shown to be required for powdery mildew susceptibility. Putative CaMBDs in the C-terminus of CsMLO1 and CsMLO2 were located at Ala 453–Arg 471 and Ala 443–Arg 461 (Appendix A), which was highly conserved throughout the *Arabidopsis* MLO family. The Ca^2+^-dependent CaM-binding motif contained Trp 419, which is known to play a significant role in CaM binding to CaM target proteins [43]. The potential functional conservation of CsMLO1 and CsMLO2 suggested that they were co-orthologs of the monocot barley HvMLO and the dicot *Arabidopsis* AtMLO2, AtMLO6, and AtMLO12 proteins.

### 2.3. Subcellular Localization and Expression Patterns of CsMLO1 and CsMLO2

A sequence analysis indicated that the proteins encoded by CsMLO1 and CsMLO2 were transmembrane proteins with seven transmembrane structures (Appendix A). To confirm this prediction, we determined the subcellular localizations of the CsMLO1 and CsMLO2 proteins, and confocal imaging showed that CsMLO1-GFP and CsMLO2-GFP fluorescence were localized in the plasma membrane, whereas the GFP protein was found in multiple subcellular compartments, including the cell membrane, cytoplasm, and nucleus (Figure 1). 

To investigate the spatial expression patterns of *CsMLO1* and *CsMLO2*, we performed a qRT-PCR analysis of the tissue-specific expression patterns in different cucumber organs of susceptible and resistant cultivars (Appendix A). *CsMLO1* exhibited the highest expression in the stem followed by the leaf, but it was weakly expressed in the cotyledon and root from the susceptible cultivar (Appendix A). However, in the resistant cultivars, *CsMLO1* was relatively highly expressed in the cotyledons and roots, followed by the leaves and stems (Appendix A). In addition, *CsMLO2* was significantly expressed in the leaf and cotyledon but was barely expressed in the stem and root from different cucumber organs of susceptible and resistant cultivars (Appendix A). Therefore, the combined subcellular localization and spatial expression patterns indicated that *CsMLO1* and *CsMLO2* are both typical MLO proteins and exhibit tissue-specific expression.

### 2.4. Response to Abiotic Stresses

We investigated the expression patterns of *CsMLO1* and *CsMLO2* genes in response to plant hormones and abiotic stimuli. The transcript levels of *CsMLO1* and *CsMLO2* were observed under abiotic stimuli in the leaves of susceptible and resistant cultivars. To determine whether *CsMLO1* and *CsMLO2* were regulated by extracellular signals, we investigated the possible involvement of *CsMLO1* and *CsMLO2* in signaling pathways related to phytohormones and CaCl_2_. The expression characteristics of the *CsMLO1* gene and the *CsMLO2* gene were distinct in the resistant cultivar (Appendix A). Compared with those of the control plants, *CsMLO1* and *CsMLO2* showed different expression patterns after exogenous treatment with H_2_O_2_, ABA, SA, MeJA and CaCl_2_. The *CsMLO1* transcript level was suppressed in resistant cultivars after treatment with exogenous ABA, CaCl_2_ and H_2_O_2_. *CsMLO2* transcription was downregulated at 24 h and 48 h after treatment with exogenous H_2_O_2_, and *CsMLO2* expression was distinctly induced and peaked at 12 h, followed by a downward trend in expression at 24 h and 48 h after treatment with exogenous CaCl_2_. But, *CsMLO2* expression was upregulated after treatment with exogenous SA. Thus, these results indicated that *CsMLO1* and *CsMLO2* might be involved in ROS, ABA, SA and calcium signaling pathway.

### 2.5. Expression Patterns of CsMLO1 and CsMLO2 in the Resistant/Susceptible Cucumber Cultivars after C. cassiicola Infection

*CsMLO1* and *CsMLO2* of cucumber were found to be involved in the response to *C. cassiicola* in previous work by iTRAQ and RNA-seq, and resistant/susceptible cucumber cultivars were also investigated for disease index by disease index analysis (Appendix A). Jinyou 38 showed resistance with a low infection index (14.85), and Xintaimici showed high susceptibility with a 72.33 index at 7 dpi (days post-inoculation). To further confirm these results, we obtained images of the cucumber cultivars in response to *C. cassiicola* under different time courses. Susceptible cucumber plants showed symptoms at 48 h after *C. cassiicola* infection, and evident symptoms were found on the leaves at 72 h and 144 h, whereas only faint symptoms were detected on the leaves of the resistant cucumber plants at 72 h and 144 h (Figure 2A). To determine whether *CsMLO1* and *CsMLO2* were affected by *C. cassiicola*, we performed RT-qPCR assays to examine the expression patterns of *CsMLO1* and *CsMLO2* at different times using resistant and susceptible cultivars (Figure 2B). A comparison of the two cultivars showed that *CsMLO1* expression was induced at 24 h and peaked at 48 h in the susceptible cultivar. In the resistant cultivar, *CsMLO1* expression was largely induced at 48 and 72 h. However, *CsMLO2* expression was decreased in the initial phase in the two cultivars and appeared to be significantly higher in the susceptible cultivar than in the resistant cultivar at 48 h. This finding suggested that the expression of *CsMLO1* and *CsMLO2* appeared to be faster in the susceptible cultivar than in the resistant cultivar when inoculated with *C. cassiicola*.

### 2.6. CsMLO1 or CsMLO2 Silencing Improves Resistance to C. cassiicola in Cucumber Cotyledon

To determine whether a loss-of-function of *CsMLO1* and *CsMLO2* affected cucumber resistance to *C. cassiicola*, we used tobacco rattle virus (TRV)-based virus-induced gene silencing (VIGS) to silence *CsMLO1*/*CsMLO2* transcripts in the susceptible cultivar Xintaimici. A 3′-terminal fragment specific to *CsMLO1*/*CsMLO2* was inserted in the vector pTRV2, the pTRV2-*CsMLO1*/*CsMLO2* recombinant plasmids (Figure 3A). Subsequently, TRV:*CsMLO1* or TRV:*CsMLO2* virus was injected in cucumber cotyledons. The chlorotic mosaic symptoms of TRV:00, TRV:*CsMLO1*, TRV*:CsMLO2* emerged in the cotyledons of transgenic cucumber plants, while no symptoms appeared in the non-injected cucumber cotyledons, suggesting that TRV successfully invaded the cucumber plants (Figure 3B). Simultaneously, efficient silencing of *CsMLO1* and *CsMLO2* was confirmed by RT-qPCR (Figure 3C). In addition, to test whether the *CsMLO2* gene was also silenced in *CsMLO1*-silenced cucumber plants, we detected *CsMLO2* expression and observed no silencing by RT-qPCR. Similarly, we detected *CsMLO1* expression in *CsMLO2*-silenced cucumber cotyledon. These results indicated that expression was almost abolished in *CsMLO1*-silenced and *CsMLO2*-silenced cucumbers.

The defense resistance of *CsMLO1*-silenced and *CsMLO2*-silenced cucumbers cotyledon was evaluated following *C. cassiicola* inoculation for 5 d in transgenic cucumbers (Figure 4). To confirm the role of candidate genes in the defense response of *C. cassiicola*, we measured the lesion size caused by the pathogen infection. Based on observations and lesion-size measurement results, we found that the sizes of the lesions formed in the cucumber leaves of the non-injected cucumber cotyledons and TRV:00-injected cucumber cotyledons were equivalent. However, the lesion area of the *CsMLO1*/*CsMLO2*-silenced cucumber cotyledons was significantly smaller than that of the non-injected cucumber cotyledons and TRV:00-injected groups. The results suggested that *CsMLO1*-silenced or *CsMLO2*-silenced enhanced disease resistance to *C. cassiicola* in cucumber.

### 2.7. Transient Overexpression of CsMLO1 or CsMLO2 in Cucumber Cotyledons Impairs Resistance to C. cassiicola

To further investigate whether the defense responses were affected by *CsMLO1*/*CsMLO2* in cucumber cotyledons, we constructed overexpression vectors containing *CsMLO1* and *CsMLO2* fused with luciferase (LUC). *Agrobacterium* carrying LUC:*CsMLO1*, LUC:*CsMLO2*, and LUC:00 was infiltrated into the cotyledons of susceptible cucumber (Figure 5). The PCR analysis showed that the introduced genes could be detected in transient overexpression plants by chimeric primers between the *CsMLO1/2* and LUC fusion genes (Figure 5A). A firefly luciferase (LUC) complementation imaging assay confirmed the success of the transient overexpression of *CsMLO1/2* in *vivo* (Figure 5B). Strong luminescence signals were detected in the *CsMLO1*-LUC and *CsMLO2*-LUC coexpression regions but not in the non-injected cucumber cotyledons. Furthermore, the RT-qPCR assay demonstrated that the expression of the *CsMLO1/CsMLO2* genes was enhanced compared with that of the LUC:00-injected cucumber cotyledons (Figure 5C). These results indicated that *CsMLO1* and *CsMLO2* have been successfully transiently overexpressed in cucumber cotyledons. The defense resistance levels of *CsMLO1*/*CsMLO2*-overexpressing cucumber cotyledons were assessed in cotyledons inoculated with *C. cassiicola* for 5 d. Increased lesion areas were observed in *CsMLO1*/*CsMLO2*-overexpressing plants compared with the non-injected and LUC:00-injected plants (Figure 6). Therefore, these results suggested that *CsMLO1*-overexpressing or *CsMLO2*-overexpressing in Xintaimici plants showed impaired resistance to *C. cassiicola.*

### 2.8. Transgenic Plants Show a Defense Response after C. cassiicola Challenge

Based on the previous analysis, we speculated that the gene expression of *CsMLO1* and *CsMLO2* was related to ROS. Detached leaves from the transient overexpression and silenced plants subjected to *C. cassiicola* were used to measure the accumulation of H_2_O_2_ and O_2_·^−^ using the 3,3′-diaminobenzidine (DAB) and nitrotetrazolium (NBT) staining methods, respectively (Figure 7). DAB staining showed that brown spots began to appear in *CsMLO1/CsMLO2*-silenced plants at 12 hpi (hours post-inoculation), which indicated that H_2_O_2_ began to accumulate compared to that in TRV:00 plants. At 24 and 48 hpi, a large area in the leaves was stained, showing that the leaves accumulated high levels of H_2_O_2_, which began to weakly decrease at 48 hpi (Figure 7A). However, at 24 hpi, brown spots began to appear in the *CsMLO1/CsMLO2*-overexpressing plants compared to the silenced plants (Figure 7B). At 48 hpi, deeply colored leaves and H_2_O_2_ accumulation were observed in *CsMLO1/CsMLO2*-overexpressing plants (Figure 7B). In the NBT staining experiment, blue spots began to appear at 12 hpi and large areas of blue began to appear in *CsMLO1/CsMLO2*-silenced plants at 24 and 48 hpi, whereas the O_2_·^−^ accumulation began to weakly decrease at 48 hpi compared with 24 hpi (Figure 7A). However, blue spots began to appear at 24 hpi, and large areas of blue began to appear in *CsMLO1/CsMLO2*-overexpressing plants at 48 hpi (Figure 7B). Thus, outbreaks of ROS in silenced plants appeared earlier than those of the transient overexpression plants following *C. cassiicola* challenge, suggesting that the silencing of *CsMLO1/CsMLO2* inhibited pathogen invasion or effectively initiated the defense responses. Moreover, histochemical staining of detached leaves at different treatment times showed lignin deposition (Figure 7). The accumulation of lignin was observed in *CsMLO1/CsMLO2*-silenced plants at 12, 24 and 48 h of *C. cassiicola* challenge (Figure 7A). Maximum lignin deposition was found in *CsMLO1/CsMLO2*-overexpressing cucumber cotyledons at 48 h following *C. cassiicola* challenge (Figure 7B). Similarly, silenced plants with *C. cassiicola* infection showed lignin deposition, and this change appeared earlier than that in the transient overexpression plants. We speculated that the strengthening of the cell wall in silenced plants following infection enhanced resistance and prevented pathogen invasion earlier than that in transient overexpression plants.

### 2.9. CsMLO1 or CsMLO2 Transgenic Plants Modulate the Expression Levels of Defense-Related Genes

To explore whether *CsMLO1/CsMLO2*-modulated expression influenced disease resistance against *C. cassiicola*, we analyzed cucumber pathogenesis-related proteins, including *PR1-1a* (AB698861), *PR2* (XM_011661051), and *PR3* (HM015248), in transgenic cucumbers (Figure 8). In our experiment, the transcript level of *PR1-1a* was only upregulated in *CsMLO1*-silenced cucumber cotyledons. The expression levels of the *PR2* and *PR3* genes were increased in *CsMLO1*-silenced and *CsMLO2*-silenced cucumber cotyledons compared with those of the TRV:00 plants. However, the transcript levels of *PR2* and *PR3* showed differing degrees of variance and declined in the transient overexpression plants. Based on the above results, we speculate that the *CsRP* might be involved in the defense pathway by *CsMLO1/CsMLO2*-induced defense pathways, whereas the *CsMLO1*-induced defense pathway might be related to the *CsRP1*-*1a* gene.

### 2.10. CsMLO1 Transient Transgenic Plants Modulate the Expression Levels of Reactive Oxygen Species (ROS)-Related Genes and Abscisic Acid (ABA) Signaling-Related Genes

The above study demonstrated the involvement of *CsMLO1* and *CsMLO2* responses to exogenous H_2_O_2_ in cucumber. Next, we investigated whether the ROS formation-related genes were affected in *CsMLO1* and *CsMLO2* transient transgenic plants (Figure 9). The transcript levels of the *CsRbohD* (*Cucsa.340760*) and *CsRbohF* (*Cucsa.107010*) genes were significantly increased in *CsMLO1*-silenced plants and decreased in the *CsMLO1*-overexpressing plants compared to those of the controls. However, *CsRbohD* and *CsRbohF* expression declined in *CsMLO2*-silenced plants (Figure 9A). Thus, *CsMLO1* silencing promoted ROS formation. To further verify these results, we examined antioxidant enzymes in *CsMLO1* and *CsMLO2* transient transgenic plants (Figure 9B). The activities of total superoxide dismutase (SOD), peroxidase (POD) and catalase (CAT) were higher in *CsMLO1*-silenced plants than in control plants but were lower in *CsMLO1*-overexpressing plants than control plants. Subsequently, excess ROS degraded polyunsaturated lipids to form malondialdehyde (MDA); the MDA level was significantly increased in *CsMLO1*-silenced plants, and opposite results were obtained in *CsMLO1*-overexpressing plants. However, the activities of these enzymes were decreased or unchanged in *CsMLO2*-transgenic plants and did not show a regular pattern. In addition, DAB and NBT staining was used to detect H_2_O_2_ and O_2_·^−^ in *CsMLO1* and *CsMLO2* transgenic plants (Figure 9C). After *CsMLO1* silencing, the H_2_O_2_ and O_2_·^−^ levels were significantly increased, resulting in brown and blue spots that were more intense than those in the TRV:00 leaves, whereas the leaf color was not obviously changed in the *CsMLO1*-overexpressing plants compared with that in the LUC:00 plants. The DAB and NBT staining in all examined *CsMLO2* transient transgenic cucumber cotyledon was similar to that of the TRV:00 and LUC:00 leaves. Based on these results, the clear signal pathways between *CsMLO1* and ROS were further confirmed. The *CsMLO1*-silenced plants triggered ROS signaling in cucumber cotyledons as a defense response to *C. cassiicola*.

As mentioned above, the transcript levels of *CsMLO1* and *CsMLO2* responded to exogenous ABA in cucumber. We next assayed the role of *CsMLO1* and *CsMLO2* in the regulation of endogenous ABA signaling-related genes in transient transgenic cucumber cotyledon (Figure 10). Compared with the control levels, the transcript levels of *CsPYL2* (JF789829) and *CsPPC2* (JN566067) were significantly increased in *CsMLO1*-silenced plants (Figure 10A) and decreased in the *CsMLO1*-overexpressing plants; however, the transcript levels of *CsPYL2* and *CsPPC2* were reduced in the *CsMLO2*-silenced and *CsMLO2*-overexpressing plants. *SnRK2.2* (JN566071) expression was almost eliminated in the *CsMLO1*-silenced cucumbers, and a decreasing pattern also appeared in the *CsMLO2*-silenced/overexpressing plants. The transcript level of *SnRK2.2* was increased in the *CsMLO1*-overexpressing cucumbers. In contrast, *CsABI5* (XM_004149176.2) gene transcription was upregulated at least 30-fold in the *CsMLO1*-silenced cucumbers compared with the TRV:00-injected cucumbers and slightly decreased in the *CsMLO1*-overexpressing cucumbers. In addition, the transcript levels of *CsABI5* were markedly lower in the *CsMLO2*-silenced/overexpressing plants than in the TRV:00/LUC:00-injected plants. Our data suggested that *CsMLO1* negatively modulated the expression of *CsPYL2*, *CsPP2C2* and *CsABI5* and positively modulated the expression of *CsSnRK2.2*. Based on these findings, we next analyzed the accumulation of endogenous ABA via ELISAs of the cotyledons of transient transgenic cucumbers (Figure 10B). To experimentally verify the above results, we assessed the accumulation of ABA, which was enhanced in the *CsMLO1*-silenced plants compared with TRV:00 plants, whereas the accumulation of ABA was reduced in the *CsMLO1*-overexpressing plants compared with LUC:00 plants. The accumulation of endogenous ABA was increased in the *CsMLO2*-silenced/overexpressing plants. Together, these results suggested that *CsMLO1* was a negative regulator of ABA-induced gene expression in the cucumber cotyledons, thus modulating the defense response to *C. cassiicola*. Although *CsMLO1* and *CsMLO2* belong to the same family and were highly similar, they may have different mechanisms of action against pathogens.

## 3. Discussion

In this study, CDS regions of two MLO genes, namely, *CsMLO1* and *CsMLO2*, were cloned and identified. A sequence analysis showed that the CsMLO1 and CsMLO2 proteins had seven typical transmembrane domains and a CaMBD, and these domain modules were similar to those of CaMLO1, AtMLO2, AtMLO6, BrMLO1 and HvMLO1 [14,44,45]. The cucumber CaMBD1 domain of CsMLO1 interacts with CsCaM3 via yeast two-hybrid, firefly luciferase (LUC) complementation and bimolecular fluorescence complementation (BiFC) experiments [46]. Plasma membrane proteins may participate in defense mechanisms against biological stress [47]. Studies of barley *mlo* mutant plants have indicated that MLO proteins play a negative regulatory role in the defense response [14,48]. Three closely related MLO co-orthologs (*AtMLO2*, *AtMLO6*, and *AtMLO12*) in clade V were mutated to achieve complete resistance to *Golovinomyces orontii* [16,49]. In our study, a phylogenetic analysis showed that a similar number of *MLO* family genes were predicted in cucumber and *CsMLO1* and *CsMLO2* were classified into the same clade as *AtMLO2*, *AtMLO6*, *AtMLO12*, *BrMLO1* and *CaMLO1*. The results indicated that loss of function of *CsMLO1* and *CsMLO2* could play a role in cucumber defense. Pepper CaMLO2 is localized in the plasma membrane and involved in the susceptibility cell-death response and bacterial and oomycete proliferation [43]. A subcellular localization analysis showed that the CsMLO1::GFP and CsMLO2::GFP fusion proteins appeared solely in the plasma membrane as fusion proteins in *Nicotiana benthamiana* leaf cells. These results were consistent with the localization of *CsMLO1* and *CsMLO2* in cucumber protoplasts [46]. Therefore, it was further demonstrated that these results were consistent with the subcellular localization results of most *MLO* disease-resistance genes, and *CsMLO1* and *CsMLO2* showed potential functions in the plasma membrane.

Traditionally, *MLO* function has been associated with susceptibility/resistance to powdery mildew (PM) disease [13]. A deletion variant (∆174) of *CsaMLO8* has lost its function as a susceptibility gene, thus leading to PM resistance in complementation of the tomato *mlo*-mutant [50]. In recent years, studies have also found that the *MLO* gene is involved in regulating other pathogens [17,18,19,20,21,22,23]. In cucumber–*C. cassiicola* interactions, the role of the *MLO* gene is unclear. Here, we found that the lesion area was significantly higher in susceptible cucumbers than in resistant cucumbers. *CsMLO1* expression was upregulated in resistant and susceptible cultivars challenged with *C. cassiicola*. Compared with the resistant cultivar at 48 h, the susceptible cultivar showed highly upregulated *CsMLO1* expression. *CsMLO2* expression was initially downregulated in the resistant and susceptible cultivars challenged with *C. cassiicola* and then upregulated at 48 h. Similarly, *CsMLO2* expression was highly upregulated in the susceptible cultivar. These results suggested that the maximum accumulation of *CsMLO1* and *CsMLO2* in the susceptible cultivar appeared earlier than that in the resistant cultivar in the defense response to *C. cassiicola*. Currently, due to the limitation of cucumber transgenic technology, research on the disease-resistance mechanism of the *MLO* gene has been hampered in cucumber. However, an experimental method for the transient agroinfiltration of cucumber cotyledons has been established [51]. In this study, we first successfully obtained transgenic cucumbers that showed transient silencing and overexpression of *CsMLO1*/*CsMLO2*. Next, a functional analysis revealed that *CsMLO1/CsMLO2*-silenced plants showed strongly enhanced resistance to *C. cassiicola*, although *CsMLO1/CsMLO2*-overexpressing plants showed evidently impaired resistance to *C. cassiicola* in the transient transgenic cucumbers. ROS accumulation and lignin deposition also occurred earlier in *CsMLO1/CsMLO2*-silenced cucumbers than *CsMLO1/CsMLO2*-overexpressing cucumbers after *C. cassiicola* infection. Wang has shown that secondary metabolism and ROS accumulation play important roles in disease resistance during cucumber-*C. cassiicola* interactions [25]. Thus, we initially concluded that *CsMLO1* and *CsMLO2* might be negative regulatory modulators involved in the cucumber defense response to *C. cassiicola*.

In *Arabidopsis*, clade V MLOs act as negative regulators of ROS signaling, which suggests that ROS signaling is a general feature of MLO proteins [30,52,53]. Normally, plants trigger ROS signaling in response to both abiotic and biotic stresses. In cucumber, *CsMLO1* and *CsMLO2* were classified into the same clade as clade V AtMLOs. Thus, cucumber *CsMLO1* and *CsMLO2* are likely associated with ROS signaling in response to *C. cassiicola*. In addition, high concentrations of ROS chemicals trigger plant HR against the invasion of pathogens [54,55]. *AtrbohD* and *AtrbohF* are associated with the production of ROS in response to ABA and pathogen infection, such as *Pseudomonas syringae* and *Hyaloperonospora arabidopsis* [31,32,56]. In our study, we found that transcript levels of *CsMLO1* were suppressed by treatment with H_2_O_2_, whereas transcript levels of *CsRbohD* and *CsRbohF* were increased in *CsMLO1*-silenced cucumbers. Meanwhile, increased antioxidant enzymes and H_2_O_2_ and O_2_^−^· accumulation were observed in *CsMLO1*-silenced cucumbers. Similar results were observed in an earlier study of the response to ROS, which was identified using publicly available gene expression data of the *mlo2-6 mlo6-2 mlo12-1* mutant [17,30]. This pattern suggested that ROS signaling might be preactivated in the *CsMLO1*-silencing cucumber to improve defense resistance to *C. cassiicola*.

The calcium signal and the reactive oxygen species signal are inseparable in plants. Ca^2+^ influx can induces the production of reactive oxygen species, which can activate Ca^2+^ influx. Cytosolic Ca^2+^ transients modulate *RbohD-* and *RbohF*-mediated ROS accumulation as well as calmodulin (CaM)-mediated defense signaling [57,58,59,60,61]. ROS can act as secondary signals in the activation of a series of downstream pathogenesis-related proteins [62]. Cucumber *CsCaM3* is a positive modulator that enhances the defense response of *C. cassiicola* infection, but *CsMLO1* is a negative modulator to enhance the defense response of cucumbers and stably interacts with CsCaM3 and transfers CsCaM3 in the cytoplasm to the plasma membrane, thereby blocking the accumulation of CsCaM3 [41]. Cucumber *CsMLO1* silencing significantly enhances the expression of reactive oxygen species (ROS)-related genes (*CsPO1*, *CsRbohD*, and *CsRbohF*), defense marker genes (*CsPR1* and *CsPR3*), *CsCaM3* and callose deposition-related gene (*CsGSL*) under *C. cassiicola* infection [41]. Additionally, the coexpression of *CsCaM3* and *CsMLO1* significantly inhibited hypersensitive cell death after *C. cassiicola* infection, suggesting that *CsMLO1* negatively regulates *CsCaM3* expression, which results in the inhibition of defense-related gene activation. In our study, the transcript level of *PR1-1a* was only increased in *CsMLO1*-silenced cucumbers. The transcript levels of *PR2* and *PR3* were significantly increased in the *CsMLO1*/*CsMLO2*-silenced cucumbers compared with the TRV:00-injected cucumbers, whereas *PR3* expression was reduced in the *CsMLO1*/*CsMLO2*-overexpressing cucumbers. These results are consistent with the results of Xue, who reported that the enhanced expression of defense-related genes improved disease resistance against *B. cinerea* in *Arabidopsis* [63]. The *CsMLO1* gene and *CsMLO2* gene belong to the same family, and they are involved in the same defense pathways against pathogens, such as active oxygen bursts and lignin deposition; however, different regulatory mechanisms are observed, such as activation of different pathogenesis-related proteins. Different expression patterns of PR proteins also indicated the possible genetic redundancy between *CsMLO1* and *CsMLO2* paralogs in the regulation of PR genes. In summary, *CsMLO1* silencing activates calcium signaling and reactive oxygen species signaling pathways, and crosstalk between Ca^2+^ signaling and ROS signals may jointly activate downstream pathogenesis-related gene expression and antioxidant enzyme activity. Negative regulatory modes of *CsMLO2* in cucumber might be related to PR protein defense against *C. cassiicola.*

In plants, complex signaling transduction pathways occur in plant–pathogen interactions [64]. The modulation of ABA signaling plays a crucial role in plant stress responses. An example of virus-induced silencing in pepper plants suggested that *CaMLO2* acts as a negative regulator of ABA signaling in drought stress responses [65]. In this study, we detected the expression of *CsPYL2*, *CsPP2C2* and *CsSnRK2.2*, which initiate ABA signal transduction, and found that *CsMLO1* negatively modulated the expression of *PYL2* and *CsPP2C2* and positively modulated the expression of *SnRK2.2*. Moreover, endogenous ABA also showed the same trend as the gene expression in *CsMLO1*-silenced/overexpressing cucumbers. Thus, we speculated that *CsMLO1* might be involved in and regulate the ABA signaling pathway. ABA is an important signaling molecule in plant–pathogen interactions [66,67,68]. In this study, a functional analysis showed that *CsMLO1* silencing significantly increased the expression of *CsABI5*, although the transcript level of *CsABI5* was inhibited in *CsMLO1*-overexpressing cucumbers. Changes in the expression levels of *CsABI5* directly demonstrated the involvement of *CsMLO1* in ABA signaling in defense responses. Furthermore, these results were relatively similar to the augmented *CsMLO1* response to *C. cassiicola*, indicating that *CsMLO1*-silencing signals mediated and enhanced the response against the pathogen. Previous evidence showed that ABA levels were correlated with resistance to pathogen stress and that *ABI5* was highly induced in a resistant line compared with a susceptible line after *S. fuliginea* infestation [69]. These results are also consistent with the response of *CsMLO1* to *C. cassiicola*, which was associated with the ABA signaling pathway. The research find that ABA stimulate the influx of calcium ions in the leaves of maize seedlings, which in turn increased the activity of plasma membrane NADPH oxidase and the production of O_2_^−^·, resulting in an increase in ROS content [70]. ABA and H_2_O_2_ can activate Ca^2+^-CaM targets and upregulate antioxidant enzyme activity [71,72,73]. Together, these findings suggest that the functional role of *CsMLO1* in the defense response to *C. cassiicola* was also correlated with ABA signaling. The *CsMLO1* silencing increased the content of endogenous ABA and upregulated the expression of related genes in cucumber leaves, and activated the ROS signal and Ca^2+^-CaM signaling pathway. However, the above assays found that the transcription level of *CsMLO2* was affected by exogenous ABA while the expression of ABA-related genes showed a downward trend compared with that of the control in *CsMLO2*-silenced/overexpressing cucumbers. These findings further suggest that although *CsMLO2* also regulated ABA signaling, whether this signaling is involved in defense against *C. cassiicola* remains unclear.

In conclusion, the cucumber *CsMLO1* and *CsMLO2* genes were identified as negative regulators of the defense response to *C. cassiicola*. Combined with the previous studies and the results of this study, it was found that *CsMLO1* as a negative regulator mainly adopted the following resistance pathways. *CsMLO1*-silenced mainly activated the expression of Ca^2+^-, ROS-, ABA- signaling-related genes, thereby improving the resistance of cucumber to *C. cassiicola* by regulating crosstalk between the three signaling; The interaction of CsMLO1 and Ca^2+^-CaM inhibited HR response mainly including ROS burst, *CaM* and pathogenesis-related gene expression. Furthermore, cucumber *CsMLO2* regulated the expression of some pathogenesis-related proteins, which influenced the defense response to *C. cassiicola.* Taken together, these studies will provide novel insights into the important roles of *CsMLO1* and *CsMLO2* in the defense responses to *C. cassiicola*.

## 4. Materials and Methods

### 4.1. Plant Materials and Treatments

The cucumber cultivars Jinyou 38, B21-a-2-1-2, B21-a-2-2-2, F10, 995, Jinyan 4 and Xintaimici were used in this study. Jinyou 38 is an important planting variety in China that is highly resistant to *C. cassiicola*. B21-a-2-2-2 (highly susceptible to *S. fuliginea*) and B21-a-2-1-2 (highly resistant to *S. fuliginea*) are two sister cucumber lines. F10 (highly resistant to *Fusarium axysporum*) and 995 (highly susceptible to *Fusarium oxysporum*) were obtained from the Liaoning Academy of Agricultural Sciences. JingYan4 is resistant to downy mildew and *C. cassiicola* in China. Xintaimici is susceptible to *C. cassiicola* and was selected for gene transformation. The cucumbers were sown in peat soil-vermiculite (1:2, *v*/*v*) under 25 °C, a 16 h light/8 h darkness cycle, and 60% relative humidity.

### 4.2. Pathogen Growth and Inoculation

*C. cassiicola* was obtained from the species conservation center of the Chinese Academy of Agricultural Sciences, and it was streaked on PDA solid medium and then stored in an incubator at 25 °C. The spore suspensions were harvested in sterile water and adjusted to a final concentration of 2 × 10^5^ sporangia/mL. The spore suspensions were sprayed onto cotyledons of transgenic cucumbers. Inoculated cucumbers were covered with plastic to maintain moisture and grown at 25 °C in a growth chamber. Then, 3-week-old cucumber seedlings of cucumber strains were inoculated at the second leaf stage with *C. cassiicola*, and control plants were inoculated with distilled water. The disease progression of corynespora leaf spot was estimated on the basis of the severity of leaf scabs as follows: Grade 0: no scabs observed; Grade 1: Less than 1/10 of leaves infected; Grade 3: 1/10–1/4 of leaves infected; Grade 5: 1/4–1/2 of leaves infected; Grade 7: 1/2–3/4 of leaf infected; and Grade 9: More than 3/4 of leaves infected. Disease index = 100 × ∑ (no. of diseased leaves of each grade × disease grade)/(total no. leaves × 9).

### 4.3. Analysis of Sequence and Phylogenetic Tree 

The nucleotide and deduced amino acid sequences of *CsMLO1* and *CsMLO2* cDNA that encode MLO homolog proteins were analyzed, and a phylogenetic tree was generated using 19 resistant proteins of known function.

### 4.4. Subcellular Localization

The RNA of the leaves was extracted using the RNAprep Pure Plant Kit (Tiangen, Beijing, China), and synthesized into cDNA using the FastQuant RT Kit (Tiangen, Beijing, China). The *CsMLO1* and *CsMLO2* cDNA regions were amplified using PrimeSTAR GXL DNA Polymerase (Takara, Dalian, China). For the PCR analysis, the following cycling parameters were used: 98 °C (10 s), 60 °C (2 min), 68 °C (30 s), 32 cycles in total. The PCR products were detected by agarose gel electrophoresis. The PCR products were introduced into pRI101-GFP vector to form two constructs with the p35S::*CsMLO1*-GFP fusion gene and p35S::*CsMLO2*-GFP fusion gene. Recombinant plasmids were transformed into *Agrobacterium tumefaciens* strain EHA 105 and then centrifuged after overnight culture. The precipitate was cultured in induction medium (10 mmol·L^−1^ ethanesulfonic acid, pH 5.7, 10 mmol·L^−1^ MgCl_2_ and 200 mmol·L^−1^ acetosyringone), harvested and diluted to OD_600_ = 0.8, and then injected into *Nicotiana benthamiana* leaves. Two days after infiltration, the epidermal cells and protoplasts were observed using a confocal laser-scanning microscope (Leica TCS SP8, Solms, Germany). The BP505-530 filter sets (excitation 488 nm, emission 505 to 530 nm) were used to detect GFP.

Protoplasts were extracted from transformed tobacco leaves. The transformed tobacco leaves were cut into 0.5–1 mm thin strips, quickly transferred to the enzymatic hydrolysate (20 mmol·L^−1^ 2-(4-Morpholino) ethanesulfonic acid (MES), 1.5% Cellulase R-10, 0.4% Macerozyme R-10, 20 mmol·L^−1^ KCL, 0.4 mol·L^−1^ mannitol, 10 mmol·L^−1^ CaCl_2_, and 0.1% bovine serum albumin (BSA), pH 5.7), and after 30 min of vacuum infiltration, shaken at 50 rpm for 6 h. The enzymatic hydrolysate was diluted with an equal amount of W5 (0.2 mol·L^−1^ MES, 1.54 mol·L^−1^ NaCl, 1 mol·L^−1^ CaCl_2_, and 0.2 mol·L^−1^ KCl, pH 5.7) and then filtered through a 100 μm nylon membrane. The collected filtrate was centrifuged at 100 × g for 2 min. The supernatant was slowly removed, and the remaining green liquid contained protoplasts. An equal amount of precooled MGG (0.2 mol·L^−1^ MES, 0.4 mol·L^−1^ Mannitol, 1.5 mol·L^−1^ MgCl_2_) was placed in the protoplasts and placed on ice for observation using a confocal laser scanning microscope.

### 4.5. Application of Plant Abiotic and Biotic Stress

The experimental materials included one-month-old cucumber seedlings (Jinyou 38 and Xintaimici). To simulate abiotic stress, the leaves of Jinyou 38 were sprayed with 10 μmol·L^−1^ H_2_O_2_, 100 μmol·L^−1^ ABA, 1 mmol·L^−1^ salicylic acid (SA), 100 μmol·L^−1^ jasmonate (MeJA) and 10 mmol·L^−1^ CaCl_2_. The control seedlings were sprayed with distilled water (H_2_O). Biotic stress, the leaves were sprayed with *C. cassiicola* inoculation. The control seedlings were sprayed with distilled water. Abiotic stress-treated cucumber leaves were collected at 12, 24, 48 h and the inoculated leaves were harvested at seven time points (i.e., 0, 6, 12, 24, 48, 72 and 144 h post-inoculation). The harvested leaves frozen in liquid nitrogen and then stored at –80 °C.

### 4.6. Quantification of Genes Expression using Reverse-Transcription Quantitative PCR (RT-qPCR) 

Total RNA was extracted from cucumber euphylla and cucumber cotyledons by RNAprep Pure Plant Kit (Tisngen biotech, Beijing, China). The primers for candidate genes were designed using QuantPrime-a flexible primer design tool for high-throughput qPCR by http://quantprime.mpimp-golm.mpg.de/. RT-qPCR was performed on a SYBR Green I 96-I system (Roche fluorescence quantitative PCR instrument, Basel). Actin gene of cucumber was used as the internal reference [74]. The relative expressions of the target genes were calculated using the 2^−ΔΔCT^ method [75]. All the primers were listed in Appendix A.

### 4.7. Histochemical Analysis 

For 3, 3’-diaminobenzidine (DAB; Sigma, St. Louis, MO, USA) and nitrotetrazolium blue chloride (NBT; Ameresco, OH, USA) to visualize H_2_O_2_ and O_2_·**^−^** [76], transient expressions development of *CsMLO1* and *CsMLO2* in cucumber cotyledons were assessed after *C. cassiicola* inoculation at the three time points (i.e., 12, 24, 48 h post-inoculation) by DAB and NBT staining. The infected cucumber cotyledons were soaked with in 1 mg/mL DAB for 8 h and infiltrated with 0.1% NBT for 5 h, boiled for 20 min in 3:1:1 ethanol/lactic acid/glycerol and then transferred to 95% ethanol at 4 °C stored. Three independent replicates were performed for each assay.

Lignin staining was performed to visualize the degree of cell wall thickening after pathogenic infection [77]. Transient expression cucumber cotyledons after *C. cassiicola* inoculation were immersed for 24 h in stationary liquid at the indicated time points. Vacuum-treated in saturated aqueous solution of chloral hydrate for 10 min at room temperature for 2 to 3 days. The leaves were stained in 1% (*w*/*v*) phloroglucinol in 92% ethanol for 10 min at room temperature. Subsequently, the tissues were mounted with HCl, and red staining was immediately monitored using Inverted microscopy (Nikon Ts2, Tokyo, Japan).

### 4.8. Enzyme Extraction and Antioxidant Enzyme Assays

Transgenic cucumber leaves were crushed into homogenate and were extracted in pre-chilled sodium phosphate buffer (pH 7.8) and 1% (*w*/*v*) polyvinylpolypyrrolidone (PVPP). The homogenate was centrifuged at 12,000× g at 4 °C for 20 min. Activities enzyme of antioxidant enzymes superoxide dismutase (SOD), peroxidase (POD), catalase (CAT) and malondialdehyde (MDA) were immediately determined in the supernatant [78,79].

### 4.9. Virus-Induced Gene Silencing (VIGS)

pTRV (tobacco rattle virus)-based VIGS was performed to knock down the *CsMLO1* gene and *CsMLO2* gene in cucumber cotyledons. A 402 bp fragment within the 3′ region of the *CsMLO1* cDNA (nucleotides 1348–1749) and a 387 bp fragment within the 3′ region of *CsMLO2* cDNA (nucleotides 1347–1725) were cloned into the pTRV2 vector (TRV:*CsMLO1* and TRV:*CsMLO2*, respectively). The gene-specific primers are listed in Appendix A.

A 10 mL culture of each *A. tumefaciens* strain to be used was grown overnight at 28 °C in YEP medium supplemented with 100 mg·L^−1^ of rifampicin and 50 mg·L^−1^ of kanamycin. Then, 200 μL of each overnight culture was inoculated into 20 mL portions of YEP medium with antibiotics as above and cultivated at 28 °C until the culture reached selected optical densities of OD_600_ = 0.8–1.0.

The induced *Agrobacterium tumefaciens* EHA105 strains carrying different pTRV2-derived vectors (pTRV2, pTRV2-*CsMLO1* and pTRV2-*CsMLO2*) were mixed with the pTRV1 *A. tumefaciens* strain EHA105. The ratio was 1:1. The samples were supplemented with 10 mmol·L^−1^ MES, 10 mmol·L^−1^ MgCl_2_, and 200 μmol·L^−1^ acetosyringone (AS) and then coinfiltrated into fully expanded cotyledons of cucumber plants (OD_600_ = 0.4 for each construct). Plants were placed in a growth room at 22 °C with a 16 h light and 8 h dark photoperiod for growth [46].

### 4.10. CsMLO-Luciferase (LUC) Fusion Overexpression Vector

The full-length *CsMLO1* and *CsMLO2* cDNA sequences were constructed using the pCAMBIA 3301 vector with luciferase (LUC), and they were under the control of the CaMV 35S promoter in cotyledons of cucumbers. The gene-specific primers are listed in Appendix A. The correct constructs were introduced into the *A. tumefaciens* strain EHA 105. *Agrobacterium*-meditated transformation with the LUC:00, LUC:*CsMLO1* gene and LUC:*CsMLO2* gene was carried out. The experimental method was the same as the above.

### 4.11. Statistical Analysis

Primer design and sequence alignment were conducted using Primer 5 software. Data are the mean ± standard deviation from three biological replicates per cultivar. Standard errors of deviation were assessed by Excel. Statistical significance was analyzed by Student’s *t*-test (*P* < 0.05 or *P* < 0.01) using SPSS software (SPSS 22.0, Nlinedown, Guangdong, China).

## Figures and Tables

**Figure 1 ijms-20-04793-f001:**
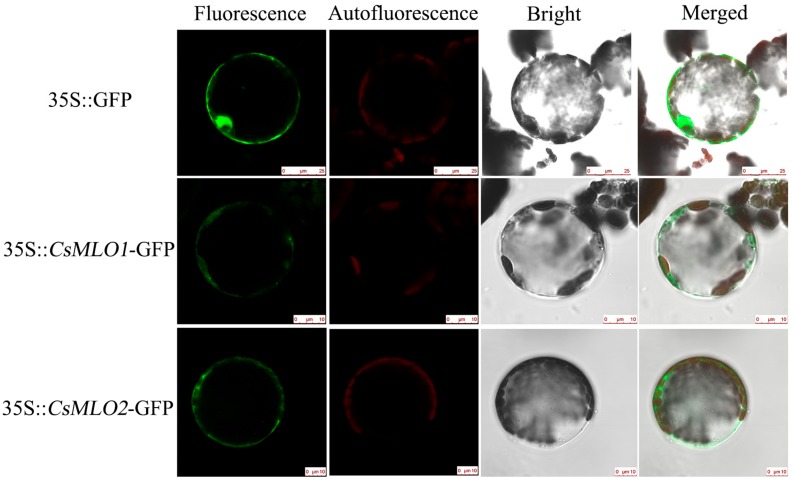
Subcellular localization of CsMLO1 and CsMLO2. Green fluorescence protein (GFP) was detected in the *N. benthamiana* protoplasts of the 35S::GFP, 35S::*CsMLO1*-GFP and 35S::*CsMLO2*-GFP constructs. CsMLO1::GFP and CsMLO2::GFP localized in the plasma membrane, GFP protein alone localized throughout the whole cells. Fluorescence, chloroplast autofluorescence, bright field and merged images were obtained using a Leica confocal microscope. Scale bars = 10 μm.

**Figure 2 ijms-20-04793-f002:**
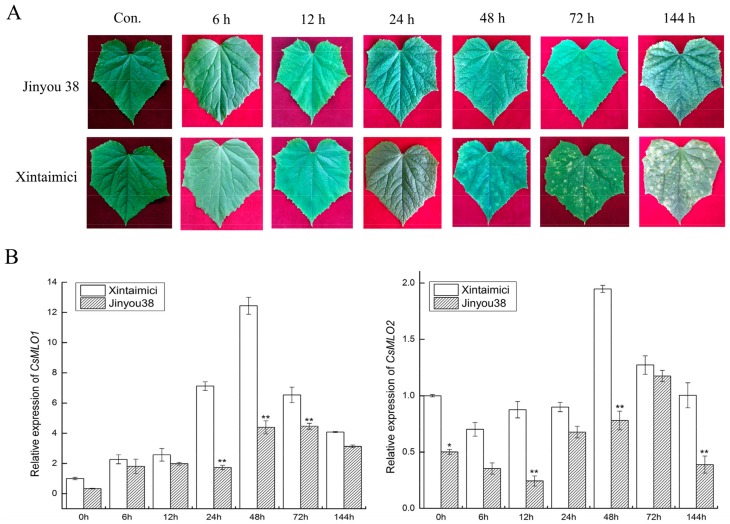
The phenotype and expression patterns of C*sMLO* in Jinyou38 and Xintaimici cultivars were inoculated with *C. cassiicola.* (**A**) Symptoms analysis in leaves of Jinyou 38 and Xintaimici cultivars after *C. cassiicola* challenge. (**B**) The transcript levels of *CsMLO1* and *CsMLO2* are shown in Jinyou 38 and Xintaimici cultivars after inoculation of *C. cassiicola*. Expression analysis of candidate genes at 0, 6, 12, 24, 48, 72 and 144 hpi (hours post-inoculation) using the 2^−ΔΔCt^ method. Data are means (± standard deviation (SD)) of three biological replicates per cultivars. The asterisks indicated a significant difference (Student’s *t*-test, **P* < 0.05 or ***P* < 0.01). Jinyou38 = high-resistant cultivar. Xintaimici = susceptible cultivar.

**Figure 3 ijms-20-04793-f003:**
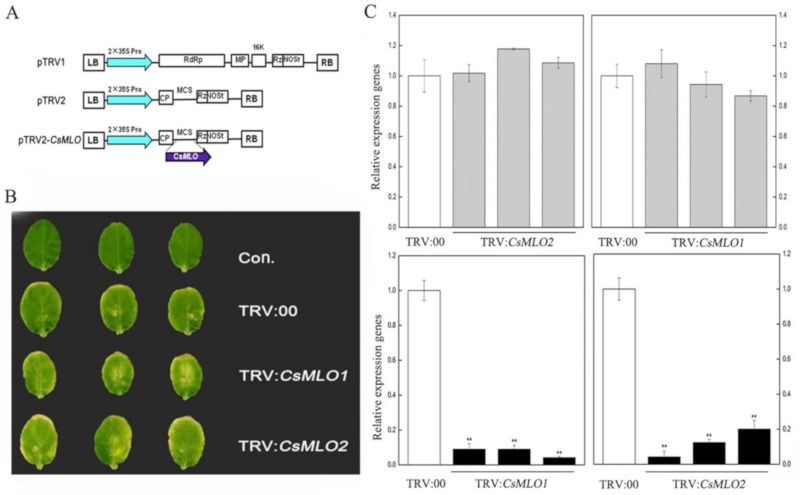
Identification of *CsMLO*-silencing cucumber plants. (**A**) Schematic of the *CsMLO1-*silenced and *CsMLO2-*silenced constructs. (**B**) Symptoms in detached cotyledons of silencing cucumber plants. Con. indicates non-injected plants. (**C**) *CsMLO1-*silenced and *CsMLO2*-silenced were identified in transgenic plants by RT-qPCR. Black bars showed the efficient silencing of *CsMLO1* and *CsMLO2* by RT-qPCR, and gray bars represented efficient silencing of *CsMLO2* in *CsMLO1-*silenced cucumber cotyledons and efficient silencing of *CsMLO1* in *CsMLO2-*silenced cucumber cotyledons. Data are means ± standard deviations from three independent experiments, and each column represents a sample containing three of cucumber cotyledons from different plants. Expression analysis of candidate genes using the 2^−ΔΔCt^ method. The asterisks indicated a significant difference (Student’s *t*-test, **P* < 0.05 or ***P* < 0.01).

**Figure 4 ijms-20-04793-f004:**
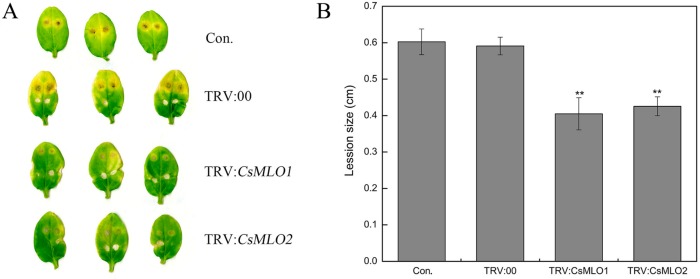
Identification of disease-resistance of silenced plants after *C. cassiicola* inoculation. (**A**) The lesion areas were observed in *CsMLO1*-silenced and *CsMLO2*-silenced cucumber cotyledons after *C. cassiicola* inoculation; (**B**) Lesion size was measured in *CsMLO1*-silenced and *CsMLO2*-silenced cucumber cotyledons after *C. cassiicola* inoculation. Data are means ± standard deviations from three biological replicates per cultivar. The asterisks indicated a significant difference (Student’s *t*-test, **P* < 0.05 or ***P* < 0.01).

**Figure 5 ijms-20-04793-f005:**
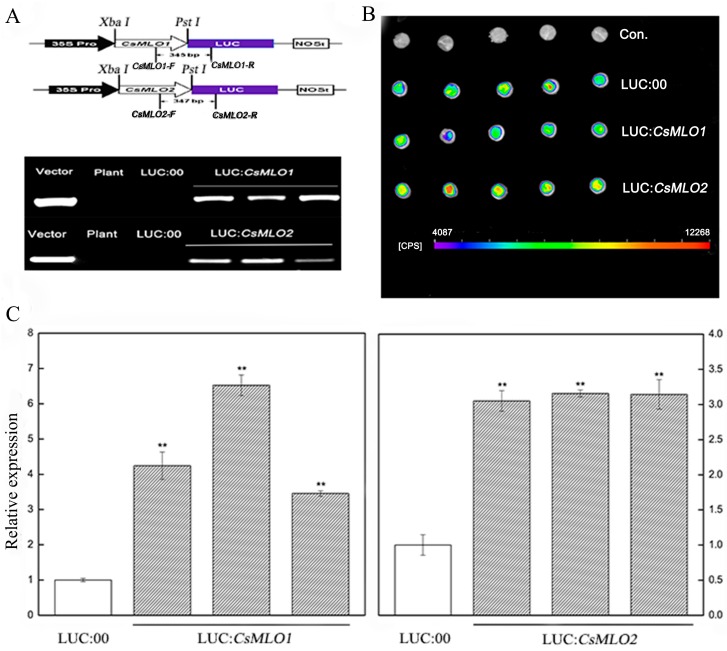
*CsMLO1*- and *CsMLO2*-transient overexpressing in cucumber cotyledons. (**A**) Schematic of the *CsMLO1-Luc* and *CsMLO2-Luc* constructs. *CsMLO1* and *CsMLO2* were between 35S Pro and Luc protein, and chimeric PCR identifications of *CsMLO1* and *CsMLO2* genetically modified of cucumber were successful. Vector, recombinant plasmid; Plant, no-transgenic cucumber; LUC:00, empty vector; LUC:*CsMLO1*, *CsMLO1*-transient overexpressing in cucumbers; LUC:*CsMLO2*, *CsMLO2*-transient overexpressing in cucumbers. (**B**) Luminescence signals were detected in the *CsMLO1*-Luc and *CsMLO2*-Luc co-expression region but not in the negative controls in cucumber cotyledons; (**C**) Transgenic plants were identified by RT-qPCR. Data are means ± standard deviations from three independent experiments, and each column represents a sample containing three of cucumber cotyledons from different plants. Expression analysis of candidate genes using the 2^−ΔΔCt^ method. The asterisks indicated a significant difference (Student’s *t*-test, **P* < 0.05 or ***P* < 0.01).

**Figure 6 ijms-20-04793-f006:**
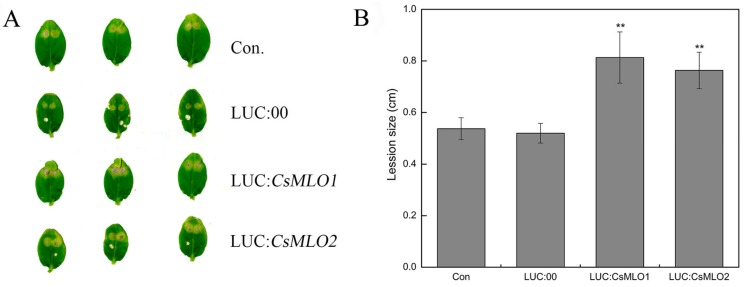
Identification of disease-resistance of transient overexpressing plants after *C. cassiicola* inoculation. (**A**) The lesion areas were observated in *CsMLO1*- and *CsMLO2*-transient overexpressing cucumber cotyledons after *C. cassiicola* inoculation. (**B**) Lesion sizes were measured in *CsMLO1*- and *CsMLO2*-transient overexpressing cucumber cotyledons after *C. cassiicola* inoculation. Data are means ± standard deviations from three biological replicates per cultivar. The asterisks indicated a significant difference (Student’s *t*-test, **P* < 0.05 or ***P* < 0.01).

**Figure 7 ijms-20-04793-f007:**
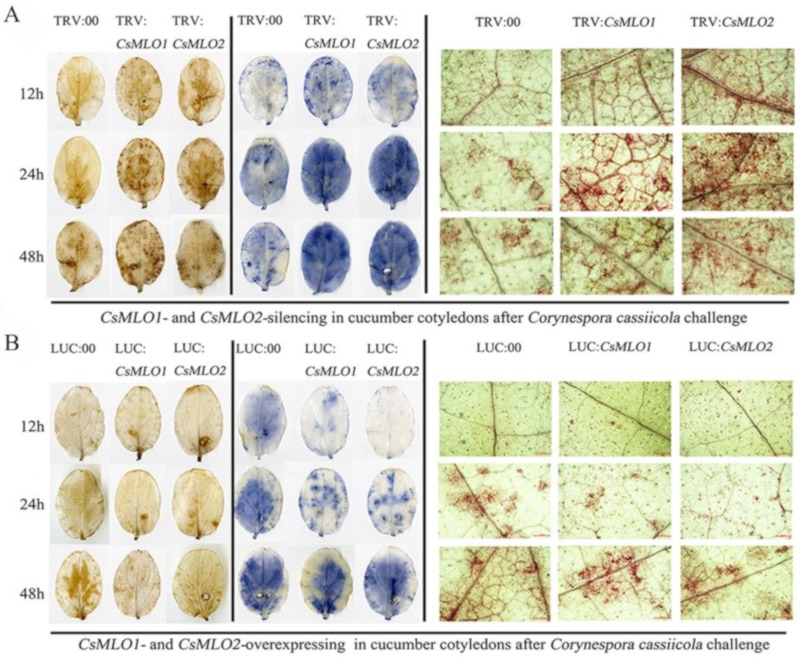
Vital staining of different times were surveyed in transgenic cucumber cotyledons after treatment with *C. cassiicola*. (**A**) The accumulation of H_2_O_2_ and O_2_·^−^ using the 3,3′-diaminobenzidine (DAB) and nitrotetrazolium (NBT) staining methods showed dye accumulation after treatment with *C. cassiicola* in silencing plants. Cell wall reinforcement showed dye accumulation after treatment with *C. cassiicola* by lignin staining. (**B**) The accumulation of H_2_O_2_ and O_2_·^−^ using the DAB and NBT staining methods showed dye accumulation after treatment with *C. cassiicola* in transient overexpressing plants. Cell-wall reinforcement showed dye accumulation after treatment with *C. cassiicola* by lignin staining. Data are means three biological replicates per cultivar. Images of lignin staining were obtained using a Nikon optical microscope. Scale bars = 500 μm.

**Figure 8 ijms-20-04793-f008:**
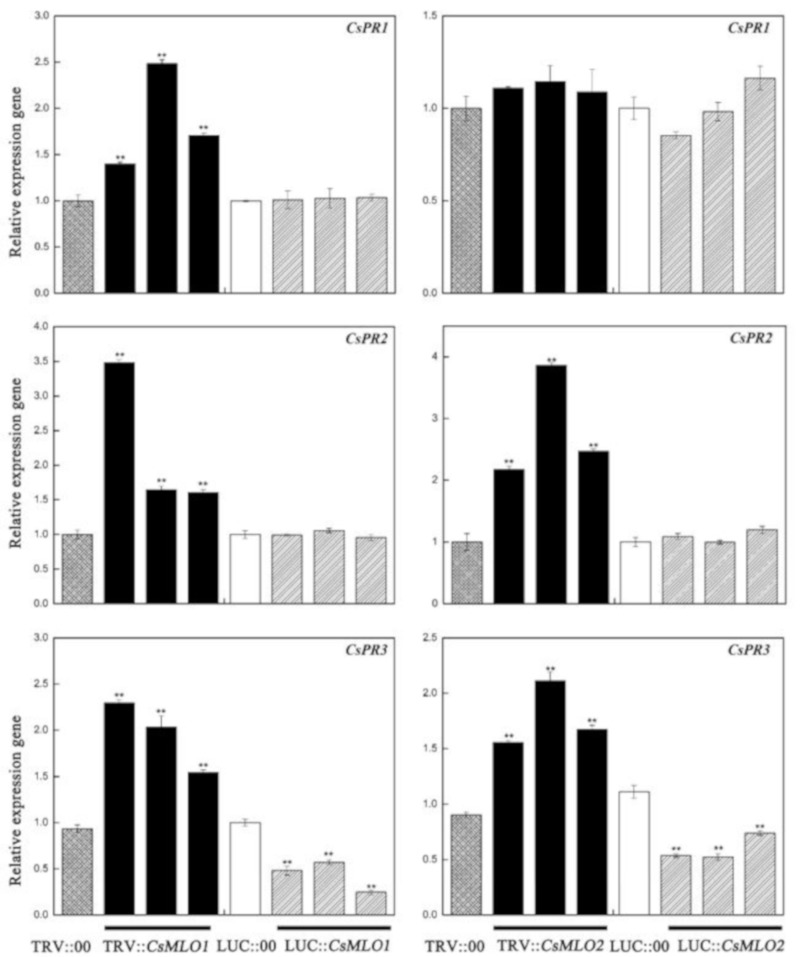
Effect of transgenic expressions of *CsMLO1* and *CsMLO2* in cucumber cotyledons on immunity induction. The different expression regulations of pathogenesis-related proteins (*CsPR1-1a*, *CsPR2* and *CsPR3*) are shown in transgenic plants. Data are means ± standard deviations from three independent experiments, and each column represents a sample containing three of cucumber cotyledons from different plants. Expression analysis of candidate genes using the 2^−ΔΔCt^ method. The asterisks indicated a significant difference (Student’s *t*-test, **P* < 0.05 or ***P* < 0.01).

**Figure 9 ijms-20-04793-f009:**
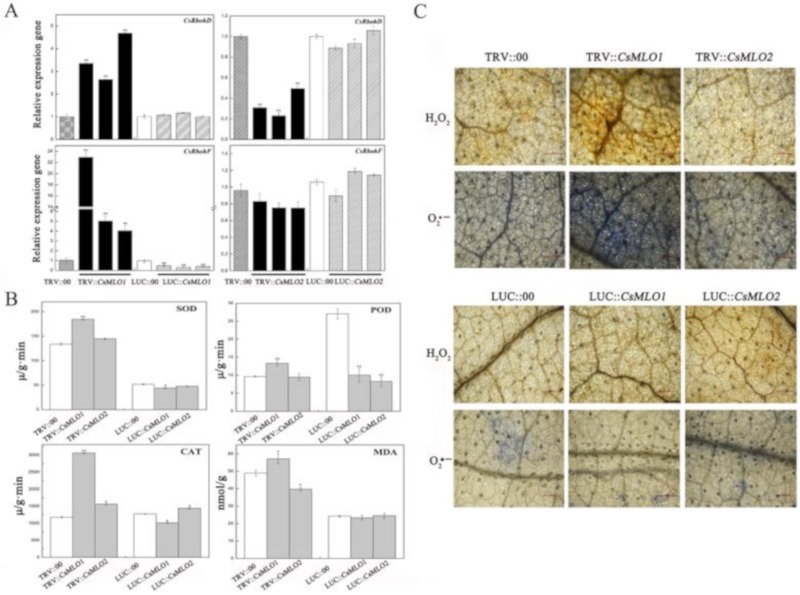
*CsMLO1* and *CsMLO2* transgenic plants modulate the ROS signaling. (**A**) The transcript level of *CsRbohD* and *CsRbohF* were detected by RT-qPCR in *CsMLO1/CsMLO2*-overexpressing and *CsMLO1/CsMLO2-*silencing cucumber cotyledons. Data are means ± standard deviations from three independent experiments, and each column represents a sample containing six of cucumber cotyledons. Expression analysis of candidate genes using the 2^−ΔΔCt^ method. The asterisks indicated a significant difference (Student’s *t*-test, **P* < 0.05 or ***P* < 0.01). (**B**) Antioxidant enzymes were examined in *CsMLO1* and *CsMLO2* transgenic plants. Data are means ± standard deviations three biological replicates per cultivar. The asterisks indicated a significant difference (Student’s *t*-test, **P* < 0.05 or ***P* < 0.01); (**C**) ROS-related (H_2_O_2_ and O_2_·^−^) staining were shown in *CsMLO1/CsMLO2*-overexpressing and *CsMLO1/CsMLO2-*silencing cucumber cotyledons. Data are means three biological replicates of per variety. Images of staining were obtained using a Nikon optical microscope. Scale bars = 500 μm.

**Figure 10 ijms-20-04793-f010:**
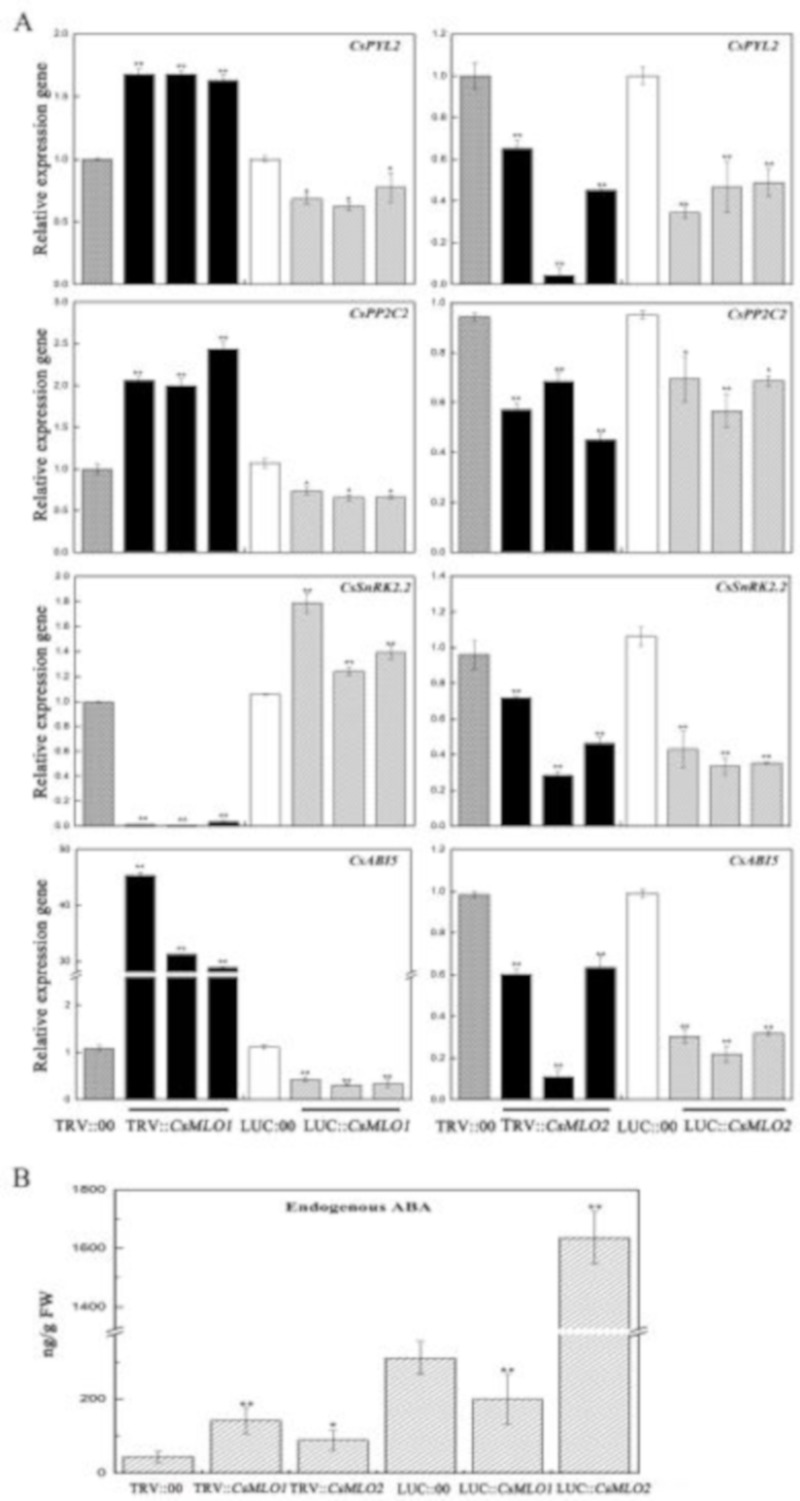
*CsMLO1* and *CsMLO2* transgenic plants modulate the ABA signaling. (**A**) ABA-related genes (*CsPYL2*, *CsPP2C2*, *CsSnRK2.2* and *CsABI5*), were observed in transgenic plants by RT-qPCR. Expression analysis of candidate genes using the 2^−ΔΔCt^ method. (**B**) The accumulation of endogenous ABA was tested in the *CsMLO1*/*CsMO2*-silencing and *CsMLO1*/*CsMO2*-overexpressing in cucumber cotyledons. Values in (**A**) and (**B**) are means ± standard deviations from three independent experiments, and each column represents a sample containing six of cucumber cotyledons. The asterisks indicated a significant difference (Student’s *t*-test, **P* < 0.05 or ***P* < 0.01).

**Table 1 ijms-20-04793-t001:** The disease index of different cucumber cultivars to *C.cassiicola.*

Material Name	Disease Index (DI)	Resistance
Jinyou 38	14.38	HR
B21-a-2-1-2	40.74	MR
B21-a-2-2-2	67.90	S
F10	41.50	MR
995	61.01	S
Jinyan 4	60.37	S
Xintaimici	70.57	S

Note: High resistance (HR), 0 < DI ≤ 15; Moderate resistance (R), 15 < DI ≤ 35; Resistance (MR), 35 < DI ≤ 55; Susceptible (S), 55 < DI ≤ 75; High susceptible (HS), DI > 75.

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
