# Peer review of "Mildew Resistance Locus O Genes CsMLO1 and CsMLO2 Are Negative Modulators of the Cucumis sativus Defense Response to Corynespora cassiicola"

_ijms, 2019, doi:10.3390/ijms20194793_

Round 1
Reviewer 1 Report
I did not see a hypothesis-driven research. The cloning of CsMLO1 and CsMLO2 appear like out of the blue. There should be a hypothesis-driven question, why did you decide to study these genes? In addition, you do not give details on how you clone both genes. In general the materials and methods used in this work are not well described, I will not be able to repeat any f the experiments that you did, because there is a lot of detail missing.I didn't understand why are you expecting chlorosis when you transiently silence the CsMLO genes. This is true with phytoene desaturase genes, which inhibit chlorophyll synthesis, but why CsMLO? could you explain? I gave more comments on the PDF file attached.

Author Response
Author's Reply to the Review Report (Reviewer 1)
Q1: I did not see a hypothesis-driven research. The cloning of CsMLO1 and CsMLO2 appear like out of the blue. There should be a hypothesis-driven question, why did you decide to study these genes?
Answer: We thank the reviewers for their comments. In our laboratory, transcriptome and iTRAQ analyses identified two CsMLOs of cucumber that are involved in the response to C. cassiicola. This phenomenon suggested that CsMLO1 and CsMLO1 may mediate resistance of cucumber to C. cassiicola. Therefore, the experiment will continue to study the function of these two genes to determine their specific resistance.
Q2: Use of strains is inadecuate. Strain is for pathogens, in the case of plants it should be used genotypes or varieties.
Answer: We accept the reviewer’s suggestion and have modified the relevant error description in the manuscript.
Q3: I don’t understand this sentence, what do you mean by the corresponding experimental materials?
Answer: We thank the reviewers for their comments. We wanted to explain: The cDNA sequences of CsMLO1 were cloned from the experimental materials of Jinyou 38 and Xintaimici, and the result showed that there was no difference in cDNA sequences of CsMLO1 between Jinyou 38 variety and Xintaimici variety. Similarly, similar cloning and alignment methods found that cDNA sequences of CsMLO2 also had no differences between Jinyou 38 variety and Xintaimici variety. We have modified the relevant description in the manuscript.
Q4: Careful with the labelling on the graphs. In this case you used letters (a,b,c) to indicate significant differences, not *
Answer: Your suggestion is correct. We have modified the relevant error description in the manuscript.
Q5: check grammar.
Answer: We apologize for this mistake. We have modified the relevant error description in the manuscript.
Q6: varieties? or only one: Jinyou Why you didn’t check for gene expression in susceptible variety Xintaimici?
Answer: We thank the reviewers for their comments. We wanted to know which hormone stress the CsMLO1 and CsMLO2 genes respond to in resistant varieties, and then explored that the CsMLO1 and CsMLO2 genes might regulated the resistance of cucumber to C. cassiicola by regulating these signaling pathways.
Q7: “disease status”.
Answer: We thank the reviewers for their comments. We have modified to ‘disease index’.
Q8: In Plant pathology, HR stands for Hypersensitive response. If the plant is resistant, you say R for resistant cultivar, variety, genotype. Otherwise is confusing. these two categories are confusing. It should be: Resistant plants (R), moderately resistant (MR) and susceptible (S). Three categories are enough. Besides, at the end you only use R and S plants.
Answer: We thank the reviewers for their comments. We have modified the relevant error description in the manuscript. Description be changed to: “High Resistant plants, 0<DI≤15; Resistant plants, 15<DI≤35; Moderately resistant, 35<DI≤55; Susceptible , 55<DI≤75”. In the current study, We used the high resistance varieties (Jinyou38).
Q9: The figure legend does not reflect the figure. F2A are leaves symptoms, and Fig.2B is gene expression.
Answer: We thank the reviewers for their comments. We have modified the relevant error description in the manuscript. Description be changed to: “The phenotype and expression pattern of CsMLO in Jinyou38 and Xintaimici varieties were inoculated with C. cassiicola”.
Q10: why if you silence MLO gene you get chlorosis? I don’t understand. You should get chlorosis when silencing Phytoene desaturase gene, or any other gene involved in chlorophyll synthesis, but I dont understand why silencing MLO will show chlorosis.
Answer: This phenotypic phenomenon is based on the results of repeated experiments. The meaning of “chlorosis” does not represent gene involvement chlorophyll synthesis. This phenotypic phenomenon of “chlorosis” and “ virus spots” shown that CsMLO1 and CsMLO2 genes successfully were silenced in cucumber cotyledons. In addition, this phenomenon also occurs in the cucumber cotyledons of TRV:00, but this phenomenon is significantly lighter than that of the cucumber cotyledons of the silenced gene.
Therefore, we have modified the relevant description in the manuscript. Description be changed to: The chlorotic mosaic symptoms of TRV:00, TRV:CsMLO1, TRV:CsMLO2 emerged in the cotyledons of transgenic cucumber plants, while no symptoms appeared in the non-injected cucumber cotyledons.
Q11: Define what is LUC here. You define it 4 sentences below.
Answer: We thank the reviewers for their comments. We have defined the LUC in the manuscript. Description be changed to: “We constructed overexpression vectors containing CsMLO1 and CsMLO2 fused with luciferase (LUC)”.
Q12: Another title should be here: Pathogen growth and inoculation.
Answer: Your suggestion is correct. We have added this title in the manuscript.
Q13: You should describe better how did you clone the cDNA of both genes.
Answer: Your suggestion is correct. We have added this description in the manuscript. Description be changed to: “The RNA of the leaves was extracted using the RNAprep Pure Plant Kit (Tiangen, Beijing, China), and synthesized into cDNA using the FastQuant RT Kit (Tiangen, Beijing, China). The CsMLO1 and CsMLO2 cDNA regions were amplified using PrimeSTAR GXL DNA Polymerase (Takara, Dalian, China). For the PCR analysis, the following cycling parameters were used: 98 °C (10 s), 60 °C (2 min), 68 °C (30 s), 32 cycles in total. The PCR products were detected by agarose gel electrophoresis. The PCR products were introduced into pRI101-GFP vector to form two constructs with the p35S::CsMLO1-GFP fusion gene and p35S::CsMLO2-GFP fusion genes”.
Q14: which time points, this is the material and methods section, it shold be properly described.
Answer: Thank you for the suggestion. We have added this description in the manuscript.
Description be changed to: “Abiotic stress-treated cucumber leaves were collected at 12 h, 24 h, 48 h and the inoculated leaves were harvested at seven time points (i.e., 0, 6, 12, 24, 48, 72 and 144 h post-inoculation). The harvested leaves frozen in liquid nitrogen and then stored at -80 °C”.
Q15: indicated where?
Answer: Thank you for the suggestion. We have added this description in the manuscript. Description be changed to: “Transient expressions development of CsMLO1 and CsMLO2 in cucumber cotyledons were assessed after C. cassiicola inoculation at the three time points (i.e., 12, 24, 48 h post-inoculation) by DAB and NBT staining”.
Q16: Acetosyringone, please use the word, then the abreviation.
Answer: Thank you for the suggestion. We have added this description in the manuscript.

Reviewer 2 Report
The manuscript reviewed aims a verification of the hypothesis on the possible involvement of the Mildew Resistance Locus O genes in mediating the non-specific resistance to target leaf spot disease of cucumber. This a well-designed complex research was performed with applying methods of phytopathology, molecular genetics and plant physiology. The results demonstrating differential modulation of CsMLO1 and CsMLO2 genes in reaction to Corynespora cassiicola by the way of regulating pathogenesis-related pathogenesis-related proteins, ROS-associated and ABA associated genes will help to highlight the mechanisms underlying interaction between the plant host Cucumis sativus and a parasite fungus. The results are illustrated with ten informative figures, one table, and eight figures and two tables in supplementary materials.
Recommendation:
It is necessary to emphasize in the Introduction section the difference between the Cca gene mediated race specific resistance to Corynespora cassiicola and non-specific resistance possibly determined by the genes of the MloA few mentioned grammar errors and faults were mentioned:
Gramma errors in the lines 44, 45; Figure S7: asterisks mentioned in the legend are absent in the figure; Figure 3 – misprint in the legend to part B: Con. instead of Col.; Figure 5B legend needs editing; Discussion section, line 472 contains incorrect information: the authors have cloned coding regions (not the whole genes); References, line 855: to add article A to the article title.
Author Response
Author's Reply to the Review Report (Reviewer 2)
Q1: It is necessary to emphasize in the Introduction section the difference between the Cca gene mediated race specific resistance to Corynespora cassiicola and non-specific resistance possibly determined by the genes of the Mlo
Answer: We thank the reviewers for their comments. We have modified the relevant description in the manuscript. the Cca gene mediated race specific resistance to C. cassiicola using marker-assisted selection (MAS) and simple sequence repeat (SSR) techniques. However, the genetic and physical distances of linked markers from the these studies are still far away from the target gene. These markers are not breeder friendly and not amendable for high-throughput genotyping.
Therefore, previous studies have found that MLO responds to cucumber TSL by transcriptome and proteomics techniques. Most plants have non-host resistance to microbial invaders, and this resistance mechanism provides broad-spectrum and strong resistance to nonadapted pathogens in plants. The mildew resistance locus O (MLO) gene in nonhost defense plays an important role in the model system of the interaction of plant with pathogen. Inducible defense responses in nonhost plants is mainly caused by the hypersensitive response (HR) of host plants, including the accumulation of ROS, the activation of pathogenesis-related genes, and localized reinforcement of the plant cell walls. Therefore, We can screen related resistance genes and study their molecular mechanisms of host resistance to cucumber TLS. The elucidating the molecular mechanism of cucumber-C. cassiicola interactions is important for establishing ideal varieties.
Q2: Gramma errors in the lines 44, 45; Figure S7: asterisks mentioned in the legend are absent in the figure; Figure 3-misprint in the legend to part B: Con. instead of Col.; Figure 5B legend needs editing; Discussion section, line 472 contains incorrect information: the authors have cloned coding regions (not the whole genes);
Answer: We thank the reviewers for their comments. We have modified the relevant error description in the manuscript.
Q3: References, line 855: to add article A to the article title.
Answer: We thank the reviewers for their comments. We have modified the relevant error in the references.
